# Impact of Thermal Radiation on Magnetohydrodynamic Unsteady Thin Film Flow of Sisko Fluid over a Stretching Surface

**Abdul Samad Khan [1],*, Yufeng Nie [1] and Zahir Shah [2]** 

[1]  Department of Applied Mathematics, School of Science, Northwestern Polytechnical University, Dongxiang Road, Chang'an District, Xi'an 710129, China; yfnie@nwpu.edu.cn
[2]  Department of Mathematics, Abdul Wali Khan University, Mardan 32300, Pakistan; zahir1987@yahoo.com
*  Correspondence: abdulsamadkhan17@mail.nwpu.edu.cn

**Abstract:** The current article discussed the heat transfer and thermal radioactive of the thin liquid flow of Sisko fluid on unsteady stretching sheet with constant magnetic field (MHD). Here the thin liquid fluid flow is assumed in two dimensions. The governing time-dependent equations of Sisko fluid are modeled and reduced to Ordinary differential equations (ODEs) by use of Similarity transformation with unsteadiness non-dimensionless parameter $St$. To solve the model problem, we used analytical and numerical techniques. The convergence of the problem has been shown numerically and graphically using Homotopy Analysis Method (HAM). The obtained numerical result shows that the HAM estimates of the structures is closed with this result. The Comparison of these two methods (HAM and numerical) has been shown graphically and numerically. The impact of the thermal radiation $Rd$ and unsteadiness parameter $St$ over thin liquid flow is discovered analytically. Moreover, to know the physical representation of the embedded parameters, like $\beta$, magnetic parameter M, stretching parameter $\xi$, and Sisko fluid parameters $\varepsilon$ have been plotted graphically and discussed.

**Keywords:** Sisko fluid; unsteady stretching sheet; thin films; MHD; HAM and numerical method

## 1. Introduction

Recently in few years it has been scrutinized that the analysis of the thin film flow has pointedly contributed in different areas like industries, engineering, and technology, etc. All problems related to thin film flow have varied applications in different fields. Some common usage of thin film associated with daily life are wire and fiber coating, extrusion of polymer and metal from die, crystal growing, plastic sheets drawing, plastic foam processing, manufacturing of plastic fluid, artificial fibers, and fluidization of reactor. Thin polymer films have abundant applications in engineering and technology. Several trade and biomedical sectors are associated with caring and functional coatings, non-fouling bio surfaces, advanced membranes, biocompatibility of medical implants, microfluidics, separations, sensors and devices, and many more. In the light of the above applications, this issue brings the attention of the researchers to improve the development of such type of study. At the initial stage the thin film flow problems discuss for viscous fluid flow and with the passage of time it turned to some Non-Newtonian Fluids. Crane [1] discussed at first time the viscid fluid motion in linear extended plate. Dandapat [2] has discussed the flow of the heat transmission investigation of the viscoelastic fluids over an extended surface. Wang [3] investigate time depending flow of a finite thin layer fluid upon a stretching plate. Ushah and Sridharan [4] used horizontal sheet for the same said problem and prolonged it to the thin partial with the analysis of heat transfer. Liu and Andersson [5]

discuss heat transmission investigation of the thin partials motion upon an unsteady extending piece. Aziz et al. [6] investigate the problem related to heat transfer of thin layer flow with thermal radioactivity. Tawade et al. [7] discussed the same problem with external magnetic field.

Furthermore, it is observed that the thin layer flow has vital roles in different fields of science. Andersson [8] is considered to be a pioneer who used the Power law model and investigated the liquid film flow viscid fluids flow upon a time dependent extending piece. Waris et al. [9] have investigated nanofluid flow with variable viscosity and thermal radioactive pass a time dependent and extending Sheet. Anderssona et al. [10] discussed the Heat transfer investigation in liquid film upon time dependent stretching plate. Chen [11,12] examined the same problem using the Power-law model. Wang et al. [13] used HAM Method to discuss the problem as examined by Chen. Saeed et al. [14] recently investigated the thin layer flow of casson Nano fluid with thermal radioactivity. The disk they took was rotating and the flow was three-dimensional. Shah et al. [15] discussed the same problem using Horizontal rotating disk. Khan et al. [16,17] studied the heat transfer investigation of the inclined magnetic field with Graphene Nanoparticles. Ullah et al. [18] studied the Brownian Motion and thermophoresis properties of the nanofluids thin layer flow of the Reiner Philippoff fluid upon an unstable stretching sheet. Shah et al. [19] investigate the thin film flow of the Williamson fluid upon an unsteady stretching surface.

Non-Newtonian fluids have abundant uses in the field of energy and technology. Plastic, food products, wall paint, greases, lubricant oil, drilling mud, etc are some common examples. Sisko fluid is also very important non-Newtonian fluids. At a very low shear rate, the Sisko flow has the same behavior as the Power-Law fluid. This property was used experimentally to fit the data of the flow of lubricating greases and also to model the flow of the whole human saliva [20]. Munir et al. studied Sisko fluid with mixed convection heat transfer. Siddiqui et al. [21,22] studied the thin film flow of Sisko fluid. Khan et al. [23] investigated the flow of boundary layer of the Sisko fluid upon a stretching surface. Molati et al. [24] discussed the MHD problem related to sisko fluid. Malik et al. [25] studied the sisko fluid with heat transfer and using convective boundary conditions.

In the past few years, the study of heat transfer and radiative flow through the stretch sheet has taken the attention from many scientists because of its large applications in engineering and industrial processes [26–31]. Rubber production, colloidal suspension, production of glass socks, metal spinning and drawing of plastic film, textile and paper production, use of geothermal energy, food processing, plasma studies, and aerodynamics are some practical examples of such flows. Radiation is often encountered in frequent engineering problems. Keeping in view its applications, Sheikholeslami et al [32–35] presented the application of radioactive nanofluid flow in different geometries. Recently some researchers studied the problems related to heat transfer. Poom et al. [36] examined the casson nanofluid with MHD radiative flow and heat source using rotating channel. Khan et al. [37] discussed the similar HMD problem as above with time dependent and thin film flow of the three fluid models Oldroyed-B, Maxwell, and Jeffry. Ishaq et al. [38] discussed the MHD effect of unsteady porous stretching surface taking Nanofluid film flow of Eyring Powell fluid. Muhammad et al. [39] discussed the MHD rotating flow upon a stretching surface with radioactively Heat absorption. Hsiao [40–43] discussed non-Newtonian fluid flow with MHD.

The purpose of this manuscript is to model and analyze thin film flow of Sisko fluid on time dependent stretching surface in the presence of the constant magnetic field (MHD). Here the thin liquid fluid flow is assumed in two dimensions. The governing time-dependent equations of Sisko fluid are modelled and reduced to ODEs by use of Similarity transformation with unsteadiness non-dimensionless parameter *St*. To solve the model problem, we used Homotopy Analysis Method-(HAM) which is one of the strongest and most time-saving methods. In addition, the heat transfer rates of thermal radiation are studied and analyzed. Shang [44] used another strong technique as Lie Algebra for solution of such type problem.

## 2. Basic Equations

The stated governing equation and heat equation are given below [23,24]:

$$divV = 0, \tag{1}$$

$$\rho \frac{dV}{dt} = -\nabla p + divS \tag{2}$$

For Sisko fluid $S$ is given as [25–28]

$$\vec{S} = \left[ a + b \left| \sqrt{\frac{1}{2} tr \binom{2}{1}} \right|^{n-1} \right]_1 \tag{3}$$

where

$$A_1 = (gradV) + (gradV)^T \tag{4}$$

wherever $a, b, n$ are the material constants which are distinct for dissimilar fluids. If we take a = 0, b = 1 and n = 0 in the Sisko fluid model then we obtained the Power-law fluid model. If we take a = 1, b = 0 and n = 1 in the Sisko fluid model then we obtained the stress–strain relationship of Newtonian fluid.

Because of two-dimensional fluid flow the velocity and the stress profile are presumed as

$$\vec{V}. = \left[ \vec{u}.(x,y), \vec{v}.(x,y), 0 \right], \ \vec{S}. = \vec{S}(x,y), T. = T(x,y) \tag{5}$$

where $\vec{u} \& \vec{v}$ are representing velocity components.

Inserting Equation (5) into Equations (1) and (2), the momentum and continuity equations reduce to the form as:

$$\vec{u}_x + \vec{v}_y = 0 \tag{6}$$

$$\rho\left(\vec{u}_t + \vec{u}\,\vec{u}_x + \vec{v}\,\vec{u}_y\right) = -p_{1_x} + a\left(\vec{u}_{xx} + \vec{u}_{yy}\right) + 2b\frac{\partial}{\partial x}\left[\vec{u}_x \left|4\left(\vec{u}_x\right)^2 + \left(\vec{u}_y + \vec{v}_x\right)^2\right|^{\frac{n-1}{2}}\right]$$
$$+ b\frac{\partial}{\partial y}\left[\left(\vec{u}_y + \vec{v}_x\right)\left|4\left(\vec{u}_x\right)^2 + \left(\vec{u}_y + \vec{v}_x\right)^2\right|^{\frac{n-1}{2}}\right] \tag{7}$$

$$\rho\left(\vec{v}_t + \vec{u}\,\vec{v}_x + \vec{v}\,\vec{v}_y\right) = -p_{1_x} + a\left(\vec{v}_{xx} + \vec{v}_{yy}\right) + b\frac{\partial}{\partial x}\left[\left(\vec{u}_y + \vec{v}_x\right)\left|4\left(\vec{u}_x\right)^2 + \left(\vec{u}_y + \vec{v}_x\right)^2\right|^{\frac{n-1}{2}}\right] +$$
$$2b\frac{\partial}{\partial y}\left[\vec{v}_y \left|4\left(\vec{u}_x\right)^2 + \left(\vec{u}_y + \vec{v}_x\right)^2\right|^{\frac{n-1}{2}}\right] \tag{8}$$

If $a = 0$, then above equations become the power law fluid and when $b = 0$, then it reduced Newtonian fluid. Introduce the dimensionless variable as:

$$u = \frac{\vec{u}}{\vec{U}}, v = \frac{\vec{v}}{\vec{U}}, t = \frac{t}{\vec{U}}, x = \frac{x}{L}, y = \frac{y}{L} \text{ and } p = \frac{p}{\rho s \vec{U}} \tag{9}$$

Equations (6) and (7) are written as

$$\frac{\partial v}{\partial t} + u\frac{\partial v}{\partial x} + v\frac{\partial v}{\partial y} = -\frac{\partial p_1}{\partial y} + \varepsilon_1\left(\frac{\partial^2 v}{\partial x^2} + \frac{\partial^2 v}{\partial y^2}\right)+$$
$$\varepsilon_2\frac{\partial}{\partial x}\left[\left(\frac{\partial u}{\partial y} + \frac{\partial v}{\partial x}\right)\left|4\left(\frac{\partial u}{\partial x}\right)^2 + \left(\frac{\partial u}{\partial y} + \frac{\partial v}{\partial x}\right)^2\right|^{\frac{n-1}{2}}\right]$$
$$+2\varepsilon_2\frac{\partial}{\partial y}\left[\frac{\partial v}{\partial y}\left|4\left(\frac{\partial u}{\partial x}\right)^2 + \left(\frac{\partial u}{\partial y} + \frac{\partial v}{\partial x}\right)^2\right|^{\frac{n-1}{2}}\right],$$

(10)

The dimensionless parameter $\varepsilon_1$ and $\varepsilon_2$ are defined as

$$\varepsilon_1 = \frac{a/\rho}{L\vec{U}} \ and \ \varepsilon_2 = \frac{b/\rho}{L\vec{U}}\left(\frac{\vec{U}}{L}\right)^{n-1}$$

(11)

The above equation after using the boundary layer approximations become as

$$\rho\left(u_t + uu_x + vu_y\right) = -p_{1_x} + au_{yy} + b\frac{\partial}{\partial y}\left(\left|u_y\right|^{n-1}u_y\right)$$

(12)

$$0 = -p_{1_y}$$

(13)

## 3. Mathematical Formulation of the Problem

Consider a time depending and electric conducting the flow of thin layer of the Sisko fluid with impact of thermal radiations during spreading surface where *x*-axis is in parallel to the slot where *y*-axis shown in the figure is orthogonal to the surface as given below in Figure 1.

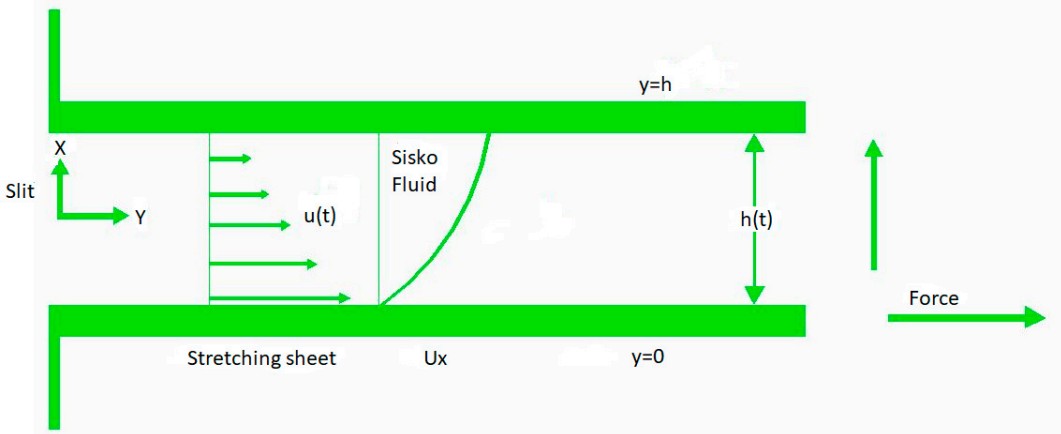

**Figure 1.** Physical Sketch of the model problem.

It is stretching flow surface also the origin is immovable because equivalent and opposed forces are acting along the X-axis. The *x*-axis is taking with stress velocity along the spreading surface.

$$U(x,t) = \frac{cx}{1 - bt}$$

(14)

In which $c$ and $b$ are constants. The *y*-axis is making a right angle to it. The term $\frac{cx^2}{v(1-bt)}$ is local Reynolds number, the surface velocity $U(x,t)$. The heat and mass transfer simultaneously here is defined as $T_s(x,t) = T_\circ - T_{ref...}\left[\frac{cx^2}{2v}\right](1-bt)^{\frac{1}{2}}$, which is surface temperature. Here $T_0$ is temperature at the slit, $T_{ref}$ is reference temperature such that $0 \le T_{ref} \le T_0$. also $T_s(x,t)$ defines temperature of the sheet, obtain from the $T_0$ at the slit in $0 \le c \le 1$.

First the slit is static with center and after an exterior power is implied to bounce the slit in the positive $x$-axis at the rate $\frac{c}{1-bt}$, where $c \in [0,1]$. An exterior transmission magnetic force is given normally to the extending sheet which is presumed to be variable type and selected as

$$B(t) = B_0(1 - bt)^{-0.5} \tag{15}$$

The fluid flow is assumed unsteady laminar and incompressible. The elementary boundary governing equations after using assumption are reduced as

$$u_x + v_y = 0 \tag{16}$$

$$u_t + uu_x + vu_y = \frac{a}{\rho}u_{yy} - \frac{b}{\rho}\frac{\partial}{\partial y}(-u_y)^n - \sigma\vec{B}^2(t)u \tag{17}$$

$$(T_t) + u(T_x) + v\left(T_y\right) = \frac{k}{\rho C_p}\left(T_{yy}\right) - \frac{1}{\rho C_p}\left(q_{r_y}\right) \tag{18}$$

Here $q_r$ approximation of Rosseland, where $q_r$ is define as [29–32],

$$q_{r_y} = -\left(\frac{4\sigma^*}{3k^*}\right)T_y^4 \tag{19}$$

Here $T$ represents the temperature fields, $\sigma^*$ is the Stefan–Boltzmann constant, $K^*$ is the mean absorption coefficient, $k$ is the thermal conductivity of the thin film. Expanding $T^4$ using Taylor's series about $T_\infty$ as below

$$T^4 = T_\infty^4 + 4T_\infty^3(T - T_\infty) + 6T_\infty^2(T - T_\infty)^2 + \dots, \tag{20}$$

Neglecting the higher order terms

$$T^4 \cong -3T_\infty^4 + 4T_\infty^3 T, \tag{21}$$

Using Equation (21) in Equation (20) we get the following

$$q_{r_y} = -\frac{16T_\infty^* \sigma^*}{3K^*}T_{yy}^4, \tag{22}$$

Using Equation (22) in Equation (18) we get the following

$$T_t + uT_x + vT_y = \frac{k}{\rho C_p}T_{yy} - \frac{1}{\rho C_p}\left(\frac{16T_\infty^* \sigma^*}{3K^*}T_{yy}^4\right), \tag{23}$$

The accompanying boundary conditions are given by

$$\vec{u} = \vec{U}_., \vec{v} = 0, T = T_s, \text{ at } y = 0, \tag{24}$$
$$\vec{u}_y = T_y = 0 \text{ at } y = h,$$

*Similarity Transformations*

The dimensionless variable $f$ and similarity variable $\eta$ for transformation as

$$.f(\eta) = \psi(x,y,t)\left(\frac{vc}{1-bt}\right)^{-\frac{1}{2}}, \eta = \sqrt{\frac{c}{v(1-bt)}}y, h(t) = \sqrt{\frac{v(1-bt)}{c}};$$
$$\theta(\eta) = T_0 - T(x,y,t)\left(\frac{bx^2}{2v(1-at)^{-\frac{3}{2}}}\left(T_{ref}\right)\right)^{-1} \tag{25}$$

Here $\psi(x_{...}, y_{...}, t_{...})$ indicate the stream function which identically satisfying Equation (16), $h(t)$ identifies the thin film thickness. The velocity components in term of stream function are obtained as

$$u = \psi_y = \left(\frac{cx}{1-bt}\right)f'(\eta), v = -\psi_x = -\left(\frac{vc}{1-bt}\right)^{\frac{1}{2}}f(\eta), \tag{26}$$

Inserting the similarity transformation Equation (25) into Equations (16)–(18) and Equation (23) fulfills the continuity Equation (16)

$$\varepsilon f''' + ff'' + n\xi(-f'')^{n-1}f''' - (f')^2 - St\left(\frac{1}{2}\eta f'' + f'\right) - Mf' = 0_{...}, \tag{27}$$

$$(1 + Rd)\theta'' - Pr\left(\frac{S}{2}(3\theta + \eta\theta') + 2f'\theta - \theta'f\right) = 0, \tag{28}$$

The boundary constrains of the problem are:

$$\begin{aligned} f'(0) &= 1, \ f(0) = 0, \theta(0) = 1, \\ f''(\beta) &= 0, \theta'(\beta) = 0, \\ f(\beta) &= \frac{S\beta}{2} \end{aligned} \tag{29}$$

The dimensionless film thickness $\beta = \sqrt{\frac{b}{v(1-\alpha t)}}h(t)$ which gives

$$h_t = -\frac{\alpha\beta}{2}\sqrt{\frac{v}{b(1-\alpha t)}} \tag{30}$$

The physical constraints after generalization are obtained as, $St = \frac{\beta}{c}$ is the non-dimensional measure of unsteadiness, $\varepsilon = \frac{a}{\rho v}$ is a Sisko fluid parameter, $\xi = \frac{b}{\rho v}\left(\left(\frac{cx}{(1-bt)\sqrt{v}}\right)^{\frac{3}{2}}\right)^{n-1}$ is stretching parameter, and $M = \frac{\sigma_f B_0^2}{b\rho_f}$ represents the magnetic, $Pr = \frac{\rho v c_p}{k} = \frac{\mu c_p}{k}$ is the Prandtl number and $Rd = \frac{4\sigma T_s^3}{kk^*}$ represent the radiation parameter.

The Skin friction is defined as

$$C_f = \frac{(S_{xy})_{y=0}}{\rho U_w^2}, \tag{31}$$

where

$$S_{xy} = \mu_0\left(a + b|u_y|^n u_y\right)y = 0 \tag{32}$$

$$C_f\sqrt{Re_x} = \varepsilon f''(0) - (-f''(0))^m \tag{33}$$

where $R_{e_x}$ is known as the local Reynolds number defined as $R_{e_x} = \frac{U_w x}{v}$. The Nusselt number is defined as $u = \frac{\delta Q_w}{\hat{k}(T-T_\delta)}$, in, which $Q_w$ is the heat flux, where $Q_w = -\hat{k}(\frac{\partial T}{\partial y})_{\eta=0}$. After the dimensionalization the $u$ is gotten the below as

$$u = -\left(1 + \frac{4}{3}Rd\right)\Theta'(0), \tag{34}$$

## 4. Application of Homotopy Analysis Method

In this section HAM is applied to Equations (27)–(29) to get an approximate analytical solution of MHD Sisko fluid flow over unsteady sheet in a following way:

$$f_0(\eta) = \eta, \theta_0(\eta) = 1, \tag{35}$$

The linear operators are

$$L_f(f) = \frac{d^3 f}{d\eta^3}, \ L_\theta(\theta) = \frac{d^2 \theta}{d\eta^2} \tag{36}$$

The above differential operators' contents are shown below as

$$
\begin{aligned}
L_f\big(\psi_1 + \psi_2 \eta + \psi_3 \eta^2\big) &= 0, \\
L_\theta(\psi_4 + \psi_5 \eta) &= 0
\end{aligned} \tag{37}
$$

where $\sum_{i=1}^{5} \psi_i$, $i = 1,2,3\ldots$ are considered as arbitrary constant.

### 4.1. Zero^{th} rder Deformation Problem

Expressing $\in [01]$ as an embedding parameter with associate parameters $\hbar_f$, and $\hbar_\theta$ where $\hbar \neq 0$. Then in case of zero order distribution the problem will be in the following form:

$$(1-)L_f\big(\hat{f}(\eta,) - f_0(\eta)\big) = h_f N_f\big(\hat{f}(\eta,)\big), \tag{38}$$

$$(1-)L_\theta\big(\hat{\theta}(\eta,) - \theta_0(\eta)\big) = \hbar_\theta N_\theta\big(\hat{f}(\eta,), \hat{\theta}(\eta,)\big),$$

The subjected boundary conditions are obtained as

$$
\begin{aligned}
f(\eta;P)\big|_{\eta=0} &= 0, \ \frac{\partial f(\eta;P)}{\partial \eta}\Big|_{\eta=0\ldots} = 1, \ \frac{\partial^2 f(\eta;P)}{\partial \eta^2}\Big|_{\eta=\beta} = 0, \\
\theta(\eta;P)\big|_{\eta=0} &= 1, \ \frac{\partial \theta(\eta;P)}{\partial \eta}\Big|_{\eta=\beta} = 0,
\end{aligned} \tag{39}
$$

The resultant nonlinear operators have been mentioned as:

$$
\begin{aligned}
N_f\big(\hat{f}(\eta;)\big) &= \varepsilon \hat{f}_{\eta\eta\eta} + n\xi\big(-f_{\eta\eta}\big)^{n-1} \hat{f}_{\eta\eta\eta} + \hat{f}_{\eta\eta}\hat{f} \\
&\quad -\big(\hat{f}_\eta\big)^2 - St\big(\hat{f}_\eta + \tfrac{\eta}{2}\hat{f}_{\eta\eta}\big) - M\hat{f}_\eta,
\end{aligned} \tag{40}
$$

$$N_{\theta\ldots}[f(\eta;), \theta(\eta;)] = (1+Rd)\theta_{\eta\eta} - \Pr\Big(\frac{S}{2}\big(3\theta + \eta\theta_\eta\big) + 2f_\eta\theta - \theta_\eta f\Big) \tag{41}$$

Expanding $\hat{f}(\eta;), \hat{\theta}(\eta;)$ in term of with use of Taylor's series expansion we get:

$$
\begin{aligned}
f_{\ldots}(\eta;P) &= f_0(\eta) + \sum_{i=1}^{\infty} f_i(\eta)P^i, \\
\theta(\eta;P) &= \theta_0(\eta) + \sum_{i=1}^{\infty} \theta_i(\eta)P^i.
\end{aligned} \tag{42}
$$

where

$$f_i(\eta) = \frac{1}{i!}\frac{\partial f(\eta;P)}{\partial \eta}\Big|_{P=0}, \ \theta_i(\eta) = \frac{1}{i!}\frac{\partial \theta(\eta;P)}{\partial \eta}\Big|_{P=0} \tag{43}$$

Here $\hbar_f$, $\hbar_\theta$ are selected in a way that the Series (43) converges at $P = 1$, switching $P = 1$ in (43), we obtain:

$$
\begin{aligned}
\hat{f}(\eta,) &= f_0(\eta) + \sum_{i=1}^{\infty} f_i(\eta), \\
\theta(\eta,) &= \theta_0(\eta) + \sum_{i=1}^{\infty} \theta_i(\eta),
\end{aligned} \tag{44}
$$

### 4.2. ith-Order Deformation Problem

Differentiating zeroth order equations $i^{th}$ time we obtained the $i^{th}$ order deformation equations with respect to . Dividing by $i!$ and then inserting $= 0$, so $i^{th}$ order deformation equations

$$
\begin{aligned}
L_f(f_i(\eta) - \xi_i f_{i-1}(\eta)) &= h_f \mathfrak{R}_i^f(\eta), \\
L_\theta(\theta_i(\eta) - \xi_i \theta_{i-1}(\eta)) &= h_\theta \mathfrak{R}_i^\theta(\eta).
\end{aligned}
\tag{45}
$$

The resultant boundary conditions are:

$$
\begin{aligned}
f_i(0) &= f_i'(0) = f_i''(\beta) = 0, \\
\theta_i(0) &= \theta_i(\beta) = 0.
\end{aligned}
\tag{46}
$$

$$
\begin{aligned}
\mathfrak{R}_i^f(\eta) &= \varepsilon f_{i-1}''' + n\xi \sum_{j=0}^{i-1}\left((-f'')^{n-1}{}_{i-1-j}f_j'''\right) - \sum_{j=0}^{i-1}\Sigma f_{i-1-j}f' \\
&\quad \sum_{j=0}^{i-1\Sigma_{i-1}'} f_{i-1-j}'f_j' - St\left(f_{i-1}' + \tfrac{\eta}{2}f_{i-1}'()\right)
\end{aligned}
\tag{47}
$$

$$
R_i^\theta(\eta) = (1+Rd)\theta_{i-1}'' - Pr\left[\left(\frac{S}{2}(3\theta_{i-1} + \eta\theta_{i-1}')\right) - \sum_{j=0}^{i-1} f_{i-1-j}\theta_j' + 2\sum_{j=0}^{i-1} f_{i-1-j}'\theta_j\right].
\tag{48}
$$

where

$$
\xi_i = \begin{cases} 1, & \text{if } P > 1 \\ 0, & \text{if } P \leq 1 \end{cases}
\tag{49}
$$

### 4.3. Convergence of Solution

After using the HAM method to calculate these solutions of the modelled function as velocity and temperature, these parameters $h_f, h_\theta$ are seen. The responsibility of the computed parameters is to regulate of convergence of the series results. In the conceivable region of $h$, $h$-curves of $f''(0)$ and $\theta'(0)$ for $12^{th}$ order approximation are plotted in Figure 2 for different values of numbers. The $h$-curves consecutively display the valid area. Table 1 values shows the numerical results of HAM solutions at dissimilar approximation. Its shows that the HAM method is fast convergent.

**Table 1.** The Homotopy Analysis Method (HAM) convergence table up to 25th order approximations when $\varepsilon = 0.5, \beta = 0.5, \xi = St = M = 0.1$.

| Approximation | $f''(0)n = 0$ | $f''(0)n = 1$ | $f''(0)n = 2$ | $f''(0)n = 3$ | $\Theta'(0)n = 2$ |
|---|---|---|---|---|---|
| 1 | 0.300000 | 0.300000 | 0.300000 | 0.300000 | −0.24761 |
| 5 | 0.489836 | 0.454647 | 0.504214 | 0.486329 | −0.214609 |
| 10 | 0.496178 | 0.457869 | 0.519242 | 1.08915 | −0.219032 |
| 15 | 0.496308 | 0.457908 | 0.520367 | 0.48611 | −0.218519 |
| 20 | 0.496311 | 0.457908 | 0.520488 | 0.485834 | −0.218509 |
| 25 | 0.496311 | 0.457908 | 0.520488 | 0.485834 | −0.218509 |

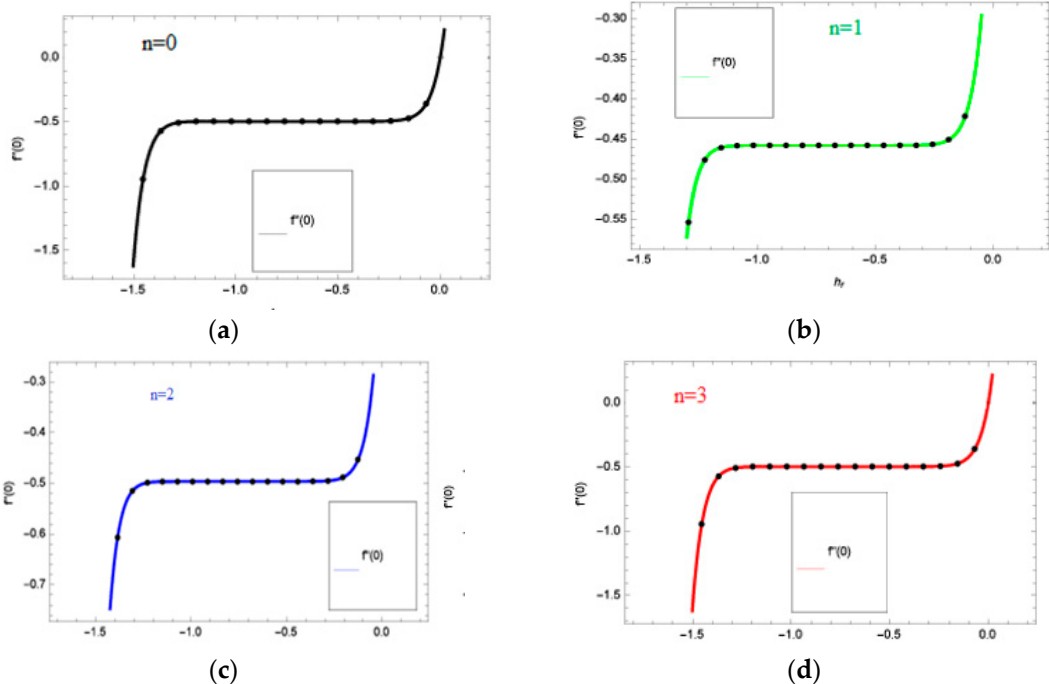

**Figure 2.** The *h*-curve graphs for velocity profile for $n = 0, 1, 2, 3$ where $\beta = 0.5, \xi = St = M = 0.1, \varepsilon = 1$.

## 5. Results and Discussion

The current work investigates the MHD and radiative flow of Sisko thin film flow having unsteady stretching sheet. The purpose of the subsection is to inspect the physical outcomes of dissimilar implanting on the velocity distributions $f(\eta)$ and temperature distribution $\Theta(\eta)$ which are described in Figures 3–12. Figure 3a–d shows the influence of unsteady constraint *St* on the velocity profile for dissimilar values of power index $n = 0, 1, 2$ and 3. Increasing *St* increase the velocity field $f(\eta)$. It is clear that varying power indices x having similar response to the time dependent parameter, that is the increase value of the power index rise the velocity distribution. The impact of the unsteadiness parameter *St* on the heat profile $\theta(\eta)$ is shown in Figure 4. It is observed that $\theta(\eta)$ directly proportional to *St*. augmented *St* rises the temperature, which in turn rises the kinetic energy of the fluid, so the fluid motion increased. It is perceived that the effect of *St* for the different value of power index $n = 0, 1, 2$ and 3 having a similar effect on heat profile $\theta(\eta)$. Figure 5a–d demonstrate the effect of the film thickness $\beta$ for dissimilar values of power index $n = 0, 1, 2$ and 3. It is perceived that the velocity profile falling down with higher values. Figure 6a–d describe the characteristics for *M* for changed values of power index $n = 0, 1, 2$ and 3. When *M* increase on the surface of the sheet during the flow, the flow rate falls, which in results decrease the velocity profiles. This significance of *M* on velocity field is due to the rise in the *M* progresses the friction force of the movement, which is called the Lorentz force. It is the fact that fluid velocity reduce in the boundary layer sheet. Figure 7 demonstrates the effect of film thickness $\beta$ on temperature profile. It is observed that the large values of film thickness $\beta$ rise the temperature, and actually the higher value of $\beta$ speed up the molecular motion of the liquids which in turn increases the internal energy and the temperature increase. The effect of stretching parameter $\xi$ for each changed values of power index $n = 0, 1, 2$ and 3 on velocity profile is shown in Figure 8a–c. In case of $n = 1$, it is clear from Figure 8a that velocity profile increase for large value of stretching parameter $\xi$. When values of power index are varied and the effect of stretching parameter $\xi$ become changed and for $n = 3$ this effect is totally opposite that is the velocity field $f(\eta)$ reduces. For $n = 0$ the stretching parameter $\xi$ becomes zero. The effect of Sisko fluid parameter $\varepsilon$ for each changed values of power index $n = 0, 1, 2$ and 3 on velocity profile is shown in Figure 9a–d. The large values of Sisko fluid parameter $\varepsilon$ rise the fluid motion, but

when the power index goes toward increase then this effect is seen changed and in case of $n = 3$ the velocity field reduces (Figure 9d). The impact of $Pr$ on $\Theta(\eta)$ is shown in Figure 10. Both temperature and concentration distributions vary in reverse form with $Pr$. When the $Pr$. number increasing it decrease the temperature distribution, and when the $Pr$ number decreases it increases the temperature distribution. Same effect of $Pr$ on concentration distribution is shown in (32). For increasing values of $Pr$ power index $n = 0, 1, 2$ and 3 thermal radiation increases rapidly. The effect of $Rd$ (thermal radiation parameter) on $\theta(\eta)$ is presented in Figure 11. When the heat transmission coefficient is minor, then the thermal radiation plays an important role in the heat transfer of comprehensive surface. By increasing thermal radiation parameter $Rd$, its show that it augments the temperature in the fluid film. Due to this rising the rate of cooling is going down. For increasing values of power index $n = 0, 1, 2$ and 3 thermal radiation increases rapidly. The HAM and numerical comparison has been displayed in in Figure 12a–d. Excellent agreement is found between HAM and numerical method.

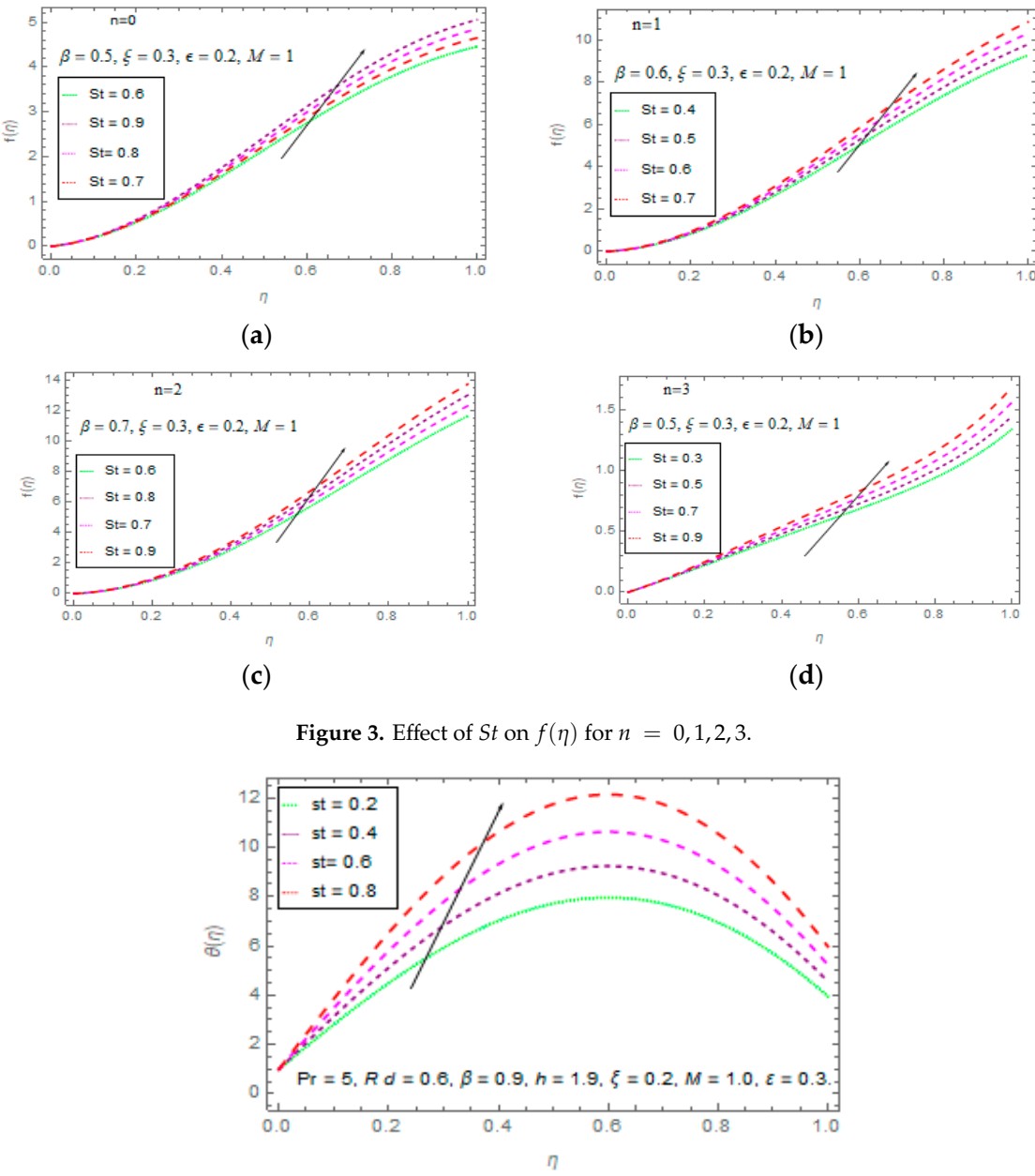

**Figure 3.** Effect of $St$ on $f(\eta)$ for $n = 0, 1, 2, 3$.

**Figure 4.** Effect of $St$ on $\theta(\eta)$.

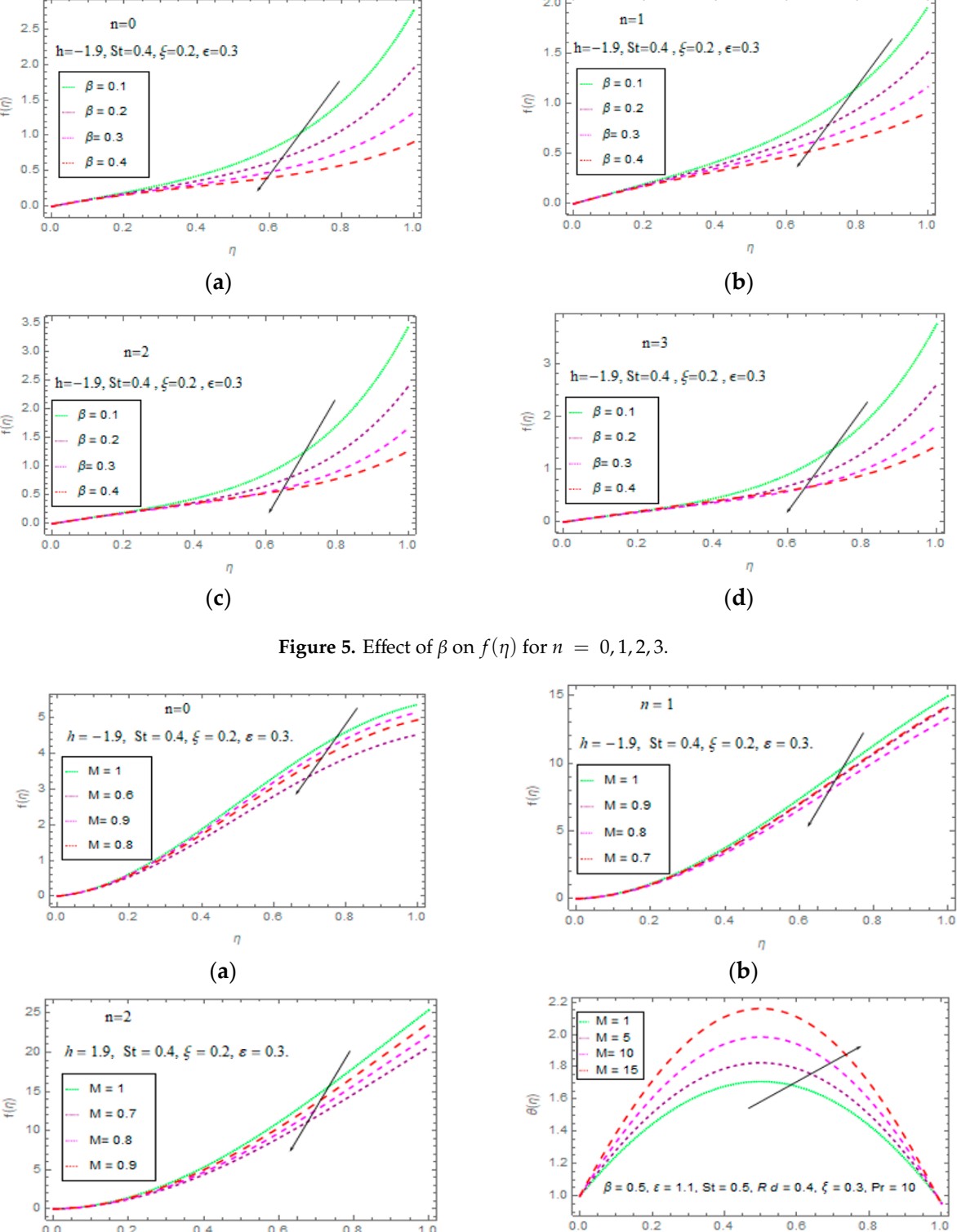

**Figure 5.** Effect of $\beta$ on $f(\eta)$ for $n = 0, 1, 2, 3$.

**Figure 6.** Effect of $M$ on $f(\eta)$ for $n = 0, 1, 2, 3$.

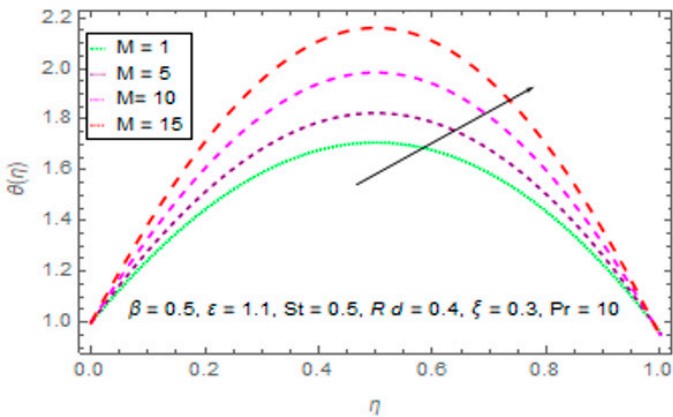

**Figure 7.** Effect of $M$ on $\theta(\eta)$.

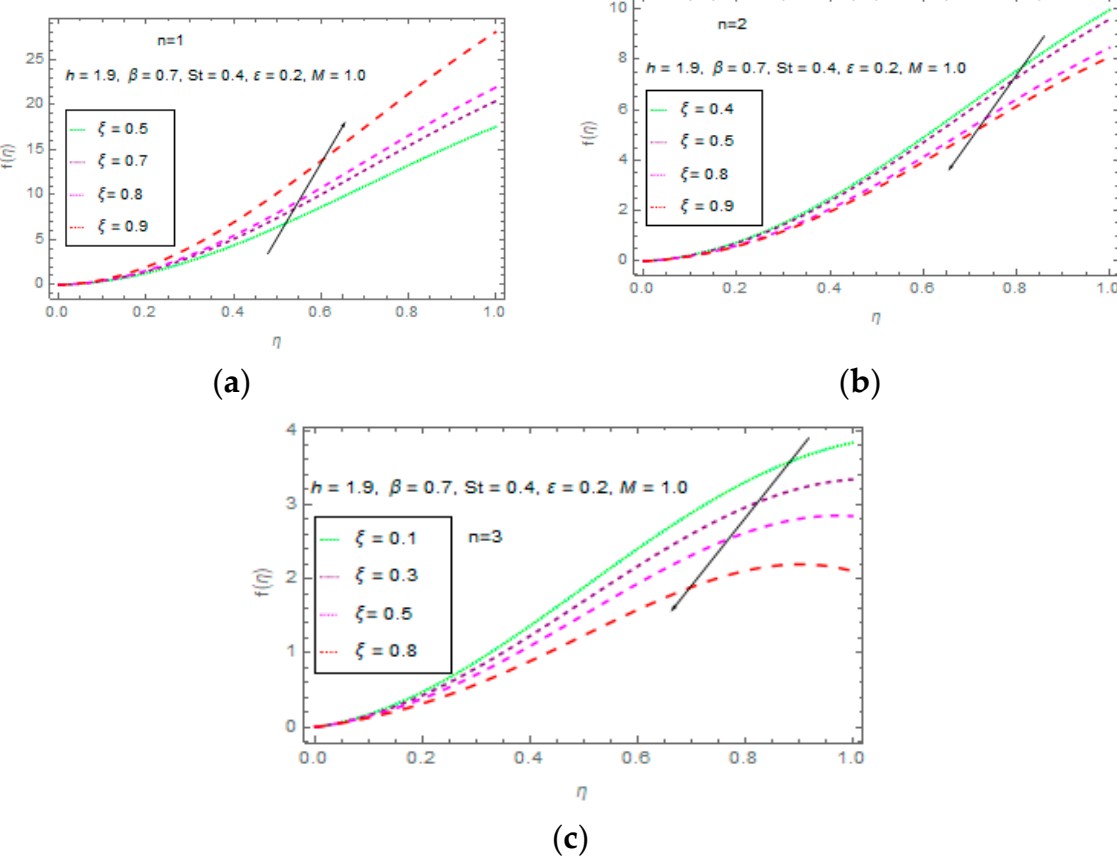

**Figure 8.** Effect of $\xi$ on $f(\eta)$ for $n = 1, 2, 3$.

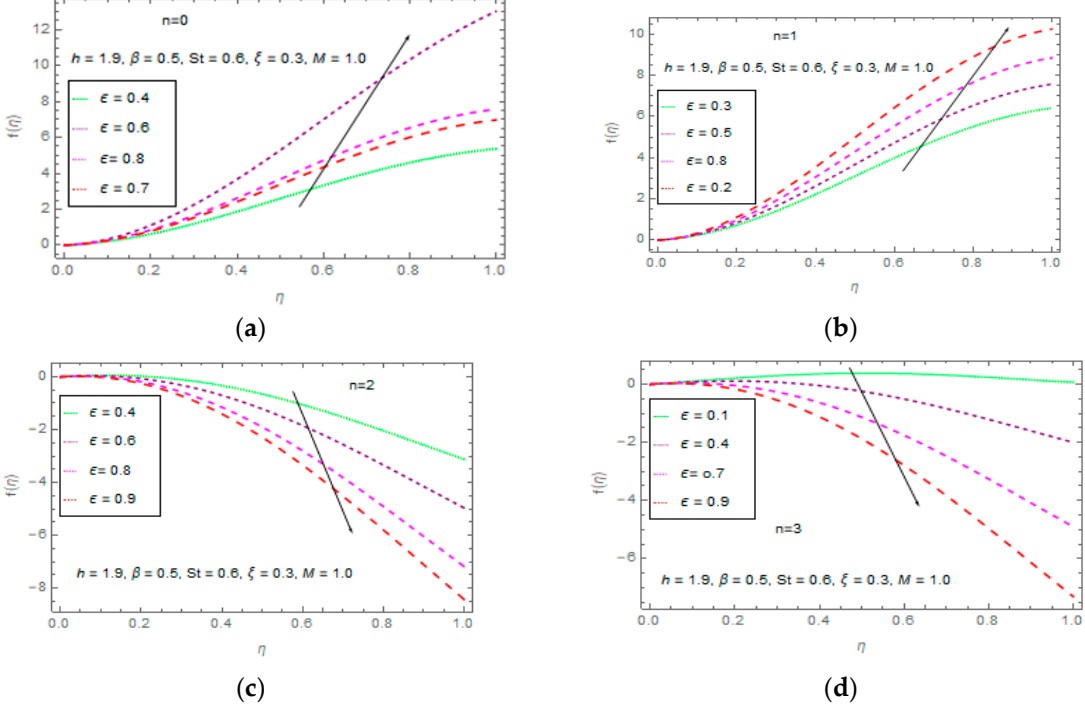

**Figure 9.** Effect of $\varepsilon$ on $f(\eta)$ for $n = 0, 1, 2, 3$.

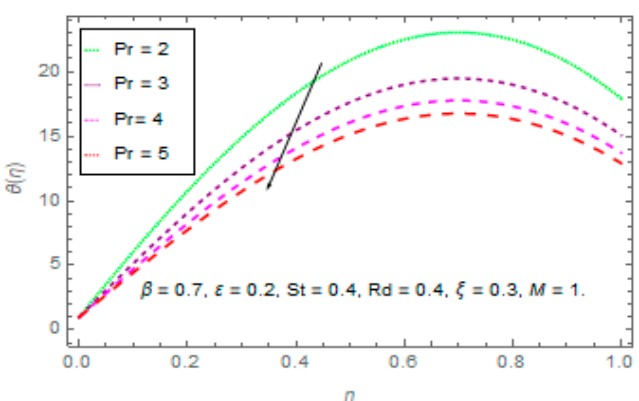

**Figure 10.** Effect of $Pr$ on $\theta(\eta)$.

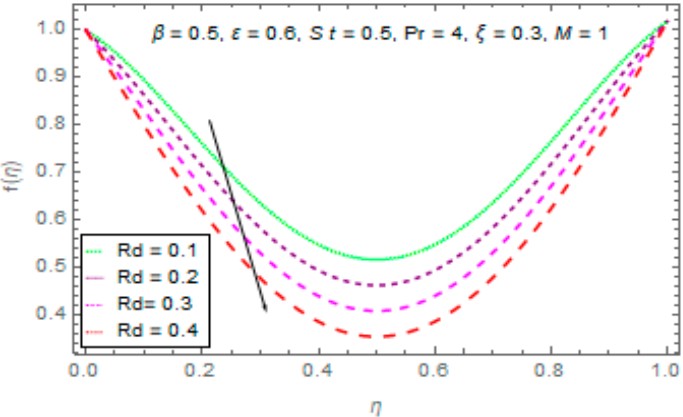

**Figure 11.** Effect of $Rd$ on $\theta(\eta)$.

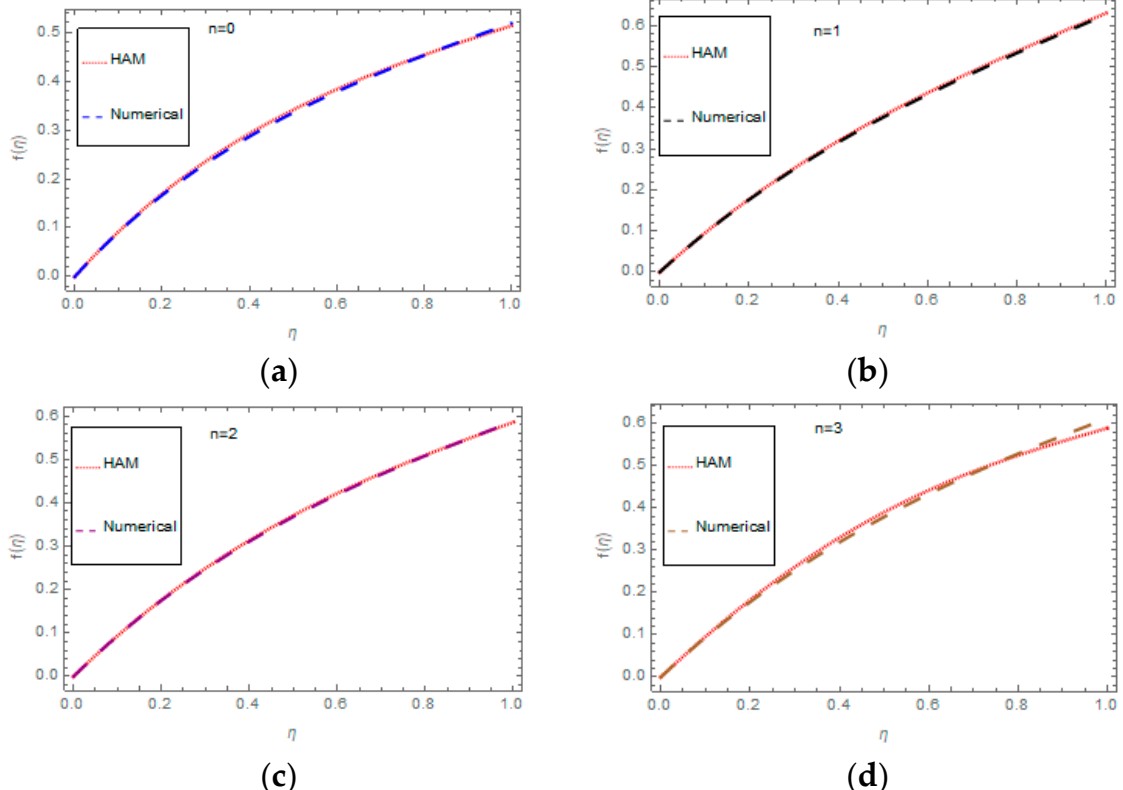

**Figure 12.** Comparison graphs between Homotopy Analysis Method (HAM) & numerical solution for velocity profiles $f(\eta)$ when $\beta = \varepsilon = 1, h = -0.6, St = M = 0.1, \xi = 0.8$.

*Table Discussion*

The modelled equations with boundary conditions) are solved analytically as well as numerically. The comparison between analytical and numerical solutions is shown graphically as well as numerically in Tables 1–5 for velocities and temperature. From these tables, an excellent agreement between HAM and Numerical (ND-Solve Techniques) are obtained. The comparison for velocity and temperature profiles for different behavior of fluid between HAM and numerical method are shown in Tables 2–6, which shows an excellent agreement with HAM solution. The numerical values of $M, k, \beta$ and $St$ on skin friction $C_f$ are given in Table 7. From this table it is clear that increasing values of $M, k$ and $\beta$ decrease $C_f$ while increasing $We$ increase skin friction. The numerical values of the surface temperature $\theta(\beta)$ for the dissimilar value of $M, Rd$ and $St$ are given in Table 8. It is observed that the increasing values of $M, Rd$ and $k$ increase the surface temperature $\theta(\beta)$, where opposite effect is found for $Pr$, that is the largest value of $Pr$ reduces the surface temperature $\theta(\beta)$. The gradient of wall temperature $\theta'(0)$ for dissimilar values of embedded parameters $Rd, \beta, Pr, S$ has been shown in Table 8. It is perceived that larger values of Rd, $\beta$ and $Pr$ fall the wall temperature and $St$ increase the wall temperature gradient $\Theta'(0)$.

**Table 2.** The relationship between Homotopy Analysis Method (HAM) and Numerical techniques for $f(\eta)$ in case of $n = 0$, when $\beta = \varepsilon = 1, St = M = 0.1, \xi = 0.8$.

| $\eta$ | HAM Solution $f(\eta)$ | Numerical Solution $f(\eta)$ | Absolute Error AE |
|---|---|---|---|
| 0 | $-2.759474 \times 10^{-21}$ | 0.0000 | $-2.759474 \times 10^{-21}$ |
| 0.1 | 0.096682 | 0.096797 | $-1.15 \times 10^{-4}$ |
| 0.2 | 0.187429 | 0.187875 | $-4.46 \times 10^{-4}$ |
| 0.3 | 0.273169 | 0.274142 | $-9.73 \times 10^{-04}$ |
| 0.4 | 0.354701 | s0.356381 | $-1.68 \times 10^{-3}$ |
| 0.5 | 0.432726 | 0.435275 | $-2.549 \times 10^{-3}$ |
| 0.6 | 0.507882 | 0.511437 | $-3.555 \times 10^{-3}$ |
| 0.7 | 0.580755 | 0.585433 | $-4.678 \times 10^{-3}$ |
| 0.8 | 0.651911 | 0.657798 | $5.887 \times 10^{-3}$ |
| 0.9 | 0.721905 | 0.729058 | $-7.153 \times 10^{-3}$ |
| 1 | 0.791305 | 0.799754 | $-8.449 \times 10^{-3}$ |

**Table 3.** Relationship between Homotopy Analysis Method (HAM) and Numerical techniques for $f(\eta)$ in case of $n = 1$, when $\beta = \varepsilon = 1, St = M = 0.1, \xi = 0.8$.

| $\eta$ | HAM Solution of $f(\eta)$ | Numerical Solution $f(\eta)$ | Absolute Error AE |
|---|---|---|---|
| 0 | $-4.44031 \times 10^{-21}$ | 0.0000 | $-2.759474 \times 10^{-21}$ |
| 0.1 | 0.097941 | 0.096797 | $-1.15 \times 10^{-4}$ |
| 0.2 | 0.192135 | 0.187875 | $-4.46 \times 10^{-4}$ |
| 0.3 | 0.273169 | 0.274142 | $-9.73 \times 10^{-4}$ |
| 0.4 | 0.354701 | 0.356381 | $-1.68 \times 10^{-3}$ |
| 0.5 | 0.432726 | 0.435275 | $-2.549 \times 10^{-3}$ |
| 0.6 | 0.507882 | 0.511437 | $-3.555 \times 10^{-3}$ |
| 0.7 | 0.580755 | 0.585433 | $-4.678 \times 10^{-3}$ |
| 0.8 | 0.651911 | 0.657798 | $5.887 \times 10^{-3}$ |
| 0.9 | 0.721905 | 0.729058 | $-7.153 \times 10^{-3}$ |
| 1 | 0.791305 | 0.799754 | $-8.449 \times 10^{-3}$ |

**Table 4.** Relationship between Homotopy Analysis Method (HAM) and ND-Solve for $f(\eta)$ in case of $n = 2$, when $\beta = \varepsilon = 1, St = M = 0.1, \xi = 0.8$.

| $\eta$ | $\theta(\eta)$ | Numerical Solution of $\theta(\eta)$ | Absolute Error AE |
|---|---|---|---|
| 0 | 1.00000 | 1.000000 | 0.000000 |
| 0.1 | 1.04865 | 1.05256 | $3.9 \times 10^{-3}$ |
| 0.2 | 1.09068 | 1.09617 | $5.5 \times 10^{-3}$ |
| 0.3 | 1.12653 | 1.13199 | $5.5 \times 10^{-3}$ |
| 0.4 | 1.15656 | 1.161 | $4.4 \times 10^{-3}$ |
| 0.5 | 1.18115 | 1.18401 | $2.9 \times 10^{-3}$ |
| 0.6 | 1.20063 | 1.20172 | $1.1 \times 10^{-3}$ |
| 0.7 | 1.21532 | 1.21472 | $6.0 \times 10^{-4}$ |
| 0.8 | 1.2255 | 1.2235 | $2.0 \times 10^{-3}$ |
| 0.9 | 1.23144 | 1.22851 | $2.9 \times 10^{-3}$ |
| 1 | 1.23337 | 1.2301 | $3.8 \times 10^{-3}$ |

**Table 5.** Association between Homotopy Analysis Method (HAM) and Numerical techniques for $\theta(\eta)$ in case of $n = 1$, when $\beta = 0.1, \varepsilon = 0.2, St = 0.5, \Pr = 0.5, \xi = 0.3, M = 1$.

| $\eta$ | $\theta(\eta)$ | Numerical Solution of $\theta(\eta)$ | Absolute Error AE |
|--------|----------------|--------------------------------------|-------------------|
| 0 | 1.0000 | 1.00000 | 0.00000 |
| 0.1 | 0.937702 | 0.9303116 | $7.4 \times 10^{-3}$ |
| 0.2 | 0.886216 | 0.875477 | $1.1 \times 10^{-2}$ |
| 0.3 | 0.844103 | 0.832878 | $1.2 \times 10^{-2}$ |
| 0.4 | 0.810135 | 0.800239 | $9.9 \times 10^{-3}$ |
| 0.5 | 0.783271 | 0.775633 | $7.6 \times 10^{-3}$ |
| 0.6 | 0.762633 | 0.757494 | $5.1 \times 10^{-3}$ |
| 0.7 | 0.74749 | 0.744597 | $2.9 \times 10^{-3}$ |
| 0.8 | 0.737235 | 0.736031 | $1.2 \times 10^{-3}$ |
| 0.9 | 0.731369 | 0.73116 | $2.1 \times 10^{-4}$ |
| 1 | 0.72949 | 0.729589 | $9.9 \times 10^{-5}$ |

**Table 6.** Association between Homotopy Analysis Method (HAM) and Numerical techniques for $\theta(\eta)$ in case of $n = 1$, when $\beta = 0.1, \varepsilon = 0.2, St = 0.5, \Pr = 0.5, \xi = 0.3, M = 1$.

| $\xi$ | $\theta(\eta)$ | Numerical Solution $\theta(\eta)$ NN | Error |
|-------|----------------|--------------------------------------|-------|
| 0 | 1.0000 | 1.00000 | 0.00000 |
| 0.1 | 1.02585 | 1.02950 | 0.00365 |
| 0.2 | 1.04897 | 1.05467 | 0.00570 |
| 0.3 | 1.06937 | 1.07579 | 0.00642 |
| 0.4 | 1.08706 | 1.09311 | 0.00605 |
| 0.5 | 1.10202 | 1.10695 | 0.00493 |
| 0.6 | 1.11426 | 1.11759 | 0.00333 |
| 0.7 | 1.12378 | 1.12533 | 0.00155 |
| 0.8 | 1.13058 | 1.13048 | 0.00010 |
| 0.9 | 1.13466 | 1.13334 | 0.00132 |
| 1 | 1.13602 | 1.13423 | 0.00179 |

**Table 7.** Coefficient of Skin friction for dissimilar values of $M, k, \beta$ and $S$.

| $M$ | $k$ | $\beta$ | $S$ | $f''(0)$ $n = 0$ | $f''(0)$ $n = 1$ | $f''(0)$ $n = 2$ | $f''(0)$ $n = 3$ |
|-----|-----|---------|-----|------------------|------------------|------------------|------------------|
| 0.1 | 0.5 | 1.0 | 1.5 | 2.6702 | 2.6702 | 3.1102 | 3.3302 |
| 0.5 |     |     |     | 1.9476 | 2.6472 | 2.9988 | 2.9488 |
| 1.0 |     |     |     | 1.7420 | 2.4421 | 2.6728 | 2.6428 |
| 1.5 | 0.1 |     |     | 2.1299 | 2.3291 | 3.4399 | 4.3399 |
|     | 0.5 |     |     | 2.3215 | 2.3223 | 3.4115 | 4.3215 |
|     | 1.0 |     |     | 2.2087 | 2.2001 | 3.2087 | 4.2687 |
|     | 1.5 | 0.1 |     | 2.6921 | 2.6957 | 4.6992 | 5.6422 |
|     |     | 0.5 |     | 2.1453 | 2.1453 | 4.5556 | 5.4456 |
|     |     | 1.0 |     | 2.3986 | 2.1986 | 3.8871 | 4.8911 |
|     |     | 1.5 | 0.1 | 2.1273 | 2.1272 | 3.0173 | 4.1273 |
|     |     |     | 0.5 | 2.3592 | 2.3392 | 3.3472 | 4.3572 |
|     |     |     | 1.0 | 2.5048 | 2.5048 | 4.5765 | 5.1048 |
|     |     |     | 1.5 | 2.9120 | 3.0020 | 5.1982 | 5.9122 |

**Table 8.** Values of $\theta(\beta)$ dissimilar values of $M, Pr, Rd$ and $S$.

| M | Pr | Rd | S | Present Result $\theta(\beta)$ | Present Result $\Theta'(0)$ |
|---|---|---|---|---|---|
| 0.0 | 0.1 | 1.0 | 0.1 | 0.2234 | 3.6823 |
| 1.0 | | | | 0.4321 | 3.5412 |
| 2.0 | | | | 0.7123 | 3.4459 |
| 5.0 | | | | 1.0230 | 3.8180 |
| 1.0 | 0.01 | | | 1.6253 | 3.4111 |
| | 0.1 | | | 1.2340 | 3.2222 |
| | 1.0 | | | 0.9882 | 5.3042 |
| | 5.0 | | | 0.5660 | 3.2914 |
| | 1.0 | 0.0 | | 0.2209 | 2.8114 |
| | | 1.0 | | 0.4320 | 1.1420 |
| | | 3.0 | | 0.6741 | 3.3714 |
| | | 5.0 | | 0.9922 | 3.1825 |
| | | 1.0 | 0.1 | 0.0112 | 2.0114 |
| | | | 0.2 | 0.2276 | 2.0005 |
| | | | 0.3 | 0.5300 | 3.4114 |
| | | | 0.4 | 0.7192 | 3.1127 |
| | | | 0.5 | 1.2005 | 2.9914 |

## 6. Conclusions

Heat transfer and thermal radiation of the thin liquid flow of Sisko fluid on time dependent stretching surface in the presence of the constant magnetic field (MHD) are investigated. Here the thin liquid fluid flow is assumed in two dimensions. The governing time-dependent equations of Sisko fluid are modeled and reduced to ODEs by use of Similarity transformation with unsteadiness non-dimensionless parameter $St$. To solve the model problem, we used analytical and numerical techniques. The convergence of the problem has been shown numerically and graphically using Homotopy Analysis Method (HAM).

The main key points are given below:

➢ In the present investigation we see that due to greater value of magnetic parameter, the velocity distribution of the thin films fluid will be decreasing.

➢ Increasing thin film thickness decreases the motion of the fluid

➢ Due to increasing radiation parameters, the Nusselt number increases.

➢ The increasing values of Pr number, raising the temperature of the surface, also caused the surface temperature to fall down for large values of unsteady parameters.

➢ The effect of the liquid film flow on the flow of Sisko fluids has been studied graphically and also shown in tables.

➢ At the end it is also summarized that due to the Lorentz's force the liquid film flow is affected.

➢ The Sisko fluid parameter increases velocity field.

➢ The effect of all parameters is shown for dissimilar values of power index $n = 0, 1, 2$ and 3.

**Author Contributions:** The 1st author "A.S.K." modeled and write the paper (Conceptualization and investigation), the 2nd author "Y.N." review, modify the paper and supervised, the 3rd author "Z.S." solve the problem (Methodology and use software).

**Funding:** This research was funded by National Natural Science Foundation of China No. [11471262].

**Conflicts of Interest:** The authors declare no conflict of interest.

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
