# Peer review of "Impact of Thermal Radiation on Magnetohydrodynamic Unsteady Thin Film Flow of Sisko Fluid over a Stretching Surface"

_processes, doi:10.3390/pr7060369_

Round 1
Reviewer 1 Report
The paper deals with an analysis of heat transfer and thermal radiation of the liquid flow of Sisko fluid. The paper needs a review by a native English to reduce the grammar typo. The paper should be accepted after minor revision improving the readability and the quality, the reviewer has highlighted some points below:
Please also avoid "lump sum references", such as¬ XXXXX [1-5]; all references should be cited with detailed and specific description. Please revise all introduction section.
The figures should be improved in terms of quality, the resolution is too low and it is difficult to read it.
Author Response
Referee’s comment (1)
Please also avoid "lump sum references", such as¬ XXXXX [1-5]; all references should be cited with detailed and specific description.
Author’s Response (1)
References are revised and corrected
Referee’s comment (2)
Please revise all introduction section.
Author’s Response (2)
Introduction are revised and edited
Referee’s comment (3)
The figures should be improved in terms of quality, the resolution is too low and it is difficult to read it.
Author’s Response (3)
All figures are edited and improved
Reviewer 2 Report
This paper studies the heat transfer and thermal radiation of the thin liquid flow of Sisko fluid on time dependent stretching surface in the presence of the constant magnetic field. The thin liquid fluid flow is assumed in two dimensions. The governing time dependent equations of Sisko fluid are modeled and reduced to ODEs by use of similarity transformation with unsteadiness non dimensionless parameter St. The convergence of the problem has been shown numerically and graphically using Homotopy Analysis Method. As far as I am concerned, the results are new. However, the presentation must be improved. I will recommend this paper for publication only if all the comments are addressed satisfactorily.
(1) The title spans 4 lines. It's a little bit long.
(2) Line 13, "The current article we discussed the...", here "we" should be deleted. Please pay attention to the grammatic issues.
(3) Line 34, "abundant application" should be "abundant applications".
(4) Line 48, "it is observe that.." should be "it is observed that.."
(5) More background on the film flow of Sisko fluid should be given in the introduction section.
(6) The motivation and novelty should be better highlighted. For a paper to be published in Processes, a certain degree of novelty is essential.
(7) Equation (3) should be explained. It is not easy to understand for those who are not in the field.
(8) I do not follow Equation (11). Some explanations are appreciated.
(9) In Equation (16), is there any particular reason to choose the power to be -0.5? What's the physical meaning behind?
(10) The homotopy analysis method presented in this work is an important analytical method. However, it should also be remarked that another tool is the Lie algebra method. See the three seminal works: A Lie algebra approach to susceptible-infected-susceptible epidemics; Analytical solution for an in-host viral infection model with time-inhomogeneous rates; Lie algebraic discussion for affinity based information diffusion in social networks.
(11) I do not understand why most of the figures are blurred. They are not up to the standard for publication. Please update these figures with original and clear ones.
(12) The conclusion should be improved by adding some future directions and open problems. This would be very helpful for the interested readers.
Author Response
Referee’s comment (1)
The title spans 4 lines. It's a little bit long.
Author’s Response (1)
Rectified
Referee’s comment (2)
Line 13, "The current article we discussed the...", here "we" should be deleted. Please pay attention to the grammatic issues.
Author’s Response (2)
Rectified
Referee’s comment (3)
Line 34, "abundant application" should be "abundant applications".
Author’s Response (3)
Rectified
Referee’s comment (4)
Line 48, "it is observe that.." should be "it is observed that.."
Author’s Response (4)
Rectified
Referee’s comment (5)
More background on the film flow of Sisko fluid should be given in the introduction section.
Author’s Response (5)
Introduction section is improved according to given suggestion
Referee’s comment (6)
The motivation and novelty should be better highlighted. For a paper to be published in Processes, a certain degree of novelty is essential.
Author’s Response (6)
Novelty of the paper are added
Referee’s comment (7)
Equation (3) should be explained. It is not easy to understand for those who are not in the field.
Author’s Response (7)
Equation (3) is corrected and typo mistakes are removed
Referee’s comment (8)
I do not follow Equation (11). Some explanations are appreciated.
Author’s Response (8)
Equation 11 is revised and corrected
Referee’s comment (9)
In Equation (16), is there any particular reason to choose the power to be -0.5? What's the physical meaning behind?
Author’s Response (9)
Here we used the Siskofluid model. Sisko fluid model, one of the various fluid models of non-Newtonian fluid, is considered for stress-strain relationship. If we take a = 0, b = 1 and n = n in the Sisko fluid model then we obtained the Power-law fluid model. If we take a = 1, b = 0 and n = 1 in the Sisko fluid model then we obtained the stress strain relationship of Newtonian fluid.
Referee’s comment (10)
The homotopy analysis method presented in this work is an important analytical method. However, it should also be remarked that another tool is the Lie algebra method. See the three seminal works: A Lie algebra approach to susceptible-infected-susceptible epidemics; Analytical solution for an in-host viral infection model with time-inhomogeneous rates; Lie algebraic discussion for affinity based information diffusion in social networks.
Author’s Response (10)
The suggested papers have been mention in the introduction part. Sorry we didn’t find the 3rd paper you mention.
Referee’s comment (11)
I do not understand why most of the figures are blurred. They are not up to the standard for publication. Please update these figures with original and clear ones.
Author’s Response (11)
Figures updated as suggested
Referee’s comment (12)
The conclusion should be improved by adding some future directions and open problems. This would be very helpful for the interested readers.
Author’s Response (12)
Conclusion improved in the revised manuscript
Reviewer 3 Report
Reviewer opinions: In this paper, the authors report the results of " Impact of Thermal Radiation on Magnetohydrodynamic Unsteady Thin Film Flow of Sisko Fluid over a Stretching Surface, Analtical and numerical study ". The work seems to be an original well done but have some minor defects existed. I enlist them below: |
Most of Figures have the boundary layer thicknessηvalue, so that Figure 2 also needed to provide the parameterζvalue. For example: Fig. 3 η value is using 1. Table 1, 6, 7, 8 are also needed the boundary layer thicknessηvalue. 3. The others related MHD non-Newtonian flow field aspect, some papers [A1-A4] studied the similar effects problems can be discussed. [A1] To Promote Radiation Electrical MHD Activation Energy Thermal Extrusion Manufacturing System Efficiency by Using Carreau-Nanofluid with Parameters Control Method, Energy, 130 (2017) 486-499 [A2] Combined Electrical MHD Heat Transfer Thermal Extrusion System Using Maxwell Fluid with Radiative and Viscous Dissipation Effects, Applied Thermal Engineering, DOI: 10.1016/j.applthermaleng.2016.08.208, (2016) [A3] Micropolar Nanofluid Flow with MHD and Viscous Dissipation Effects Towards a Stretching Sheet with Multimedia Feature, International Journal of Heat and Mass Transfer, 112 (2017) 983–990 [A4] Stagnation Electrical MHD Nanofluid Mixed Convection with Slip Boundary on a Stretching Sheet, Applied Thermal Engineering, doi:10.1016/j.applthermaleng.2015.12.138, 98 (2016) 850–861
From above, the paper still need to improve and should be revised by this time.
|
Author Response
Referee’s comment (1)
Most of Figures have the boundary layer thicknessηvalue, so that Figure 2 also needed to provide the parameterζvalue. For example: Fig. 3 η value is using 1.
Author’s Response (1)
The manuscript is edited and figures are revised and corrected
Referee’s comment (2)
Table 1, 6, 7, 8 are also needed the boundary layer thicknessηvalue.
Author’s Response (2)
All such table are corrected according given Suggestions.
Referee’s comment (3)
The others related MHD non-Newtonian flow field aspect, some papers
[A1-A4] studied the similar effects problems can be discussed.
[A1] To Promote Radiation Electrical MHD Activation Energy Thermal Extrusion Manufacturing System Efficiency by Using Carreau-Nanofluid with Parameters Control Method, Energy, 130 (2017) 486-499
[A2] Combined Electrica MHD Heat Transfer Thermal Extrusion System Using Maxwell Fluid with Radiative and Viscous Dissipation Effects, Applied Thermal Engineering, DOI: 10.1016/j.applthermaleng.2016.08.208, (2016)
[A3] Micropolar Nanofluid Flow with MHD and Viscous Dissipation Effects Towards a Stretching Sheet with Multimedia Feature, International Journal of Heat and Mass Transfer, 112 (2017) 983–990
[A4] Stagnation Electrical MHD Nanofluid Mixed Convection with Slip Boundary on a Stretching Sheet, Applied Thermal Engineering, doi:10.1016/j.applthermaleng.2015.12.138, 98 (2016) 850–861
Author’s Response (3)
The suggested papers have been mentioned in the revised manuscript.
Referee’s comment (4)
From above, the paper still need to improve and should be revised by this time.
Author’s Response (4)
The manuscript is edited. The manuscript has been revised thoroughly and all the grammatical mistakes are corrected to the level best. These mistakes have been removed in the revised manuscript.
Round 2
Reviewer 2 Report
I appreciate the author's remarkable turnaround time. The paper is in good order. Therefore, I recommend it for publication.